# A Trans and Queer Discursive Approach to Gender Diversity and Misgendering in the Transgender and Gender Diverse Population: Queering a Study for ICD-11

**DOI:** 10.3390/ijerph22020178

**Published:** 2025-01-28

**Authors:** Anna Baleige, Frédéric Denis

**Affiliations:** 1EA 75-05 Éducation Éthique Santé, Faculté de Médecine, Université François-Rabelais Tours, 2 boulevard Tonnellé, 37044 Tours, France; frederic.denis@univ-tours.fr; 2Service d’odontologie, Centre Hospitalier Universitaire de Tours, 2 boulevard Tonnellé, 37044 Tours, France

**Keywords:** transgender persons, sexual and gender minorities, terminology as topic, knowledge, community-based participatory research

## Abstract

Producing knowledge about transgender and gender-diverse (TGD) individuals is a core public health strategy challenge. Yet several systemic limitations arise, notably the exclusion or exploitation of TGD individuals by research systems reproducing systemic discrimination by embedding social norms as self-evident facts of nature. This is particularly worrying in biomedical research, and contributes to the invisibilization of participants’ gender diversity. This trans research illustrates methodological challenges through queering an earlier study by focusing on misgendering as a discursive element. We based our work on discursive materials reported by TGD participants in an ICD-11 study on gender incongruence. We used network analyses to illustrate potential differences between declared gender identity and discourse practices. Our results highlight a gap between declared gender identity and discourse practices, bringing the number of non-binary participants in the sample from 15 (20.8%) to 36 (50.0%). Moreover, misgendering and the use of derogatory terms are more common toward gender-diverse individuals. Sexual orientation shows a similar trend. This study reveals the reproduction of social norms within research processes and medical knowledge, as well as how, from an individual perspective, their non-compliance seems to be a key factor in TGD individuals’ experience. By providing this simple methodological example, we hope to promote better integration of gender and its various dimensions into biomedical and public health research.

## 1. Introduction

Promoting health among transgender and gender-diverse people (TGD) has become a central concern of the health policies affecting them. A growing body of scientific literature, particularly from a global health perspective, points to major health inequalities compared with the general population [1]. Discrimination is now argued as the main source of these inequalities, both in scientific literature [1,2,3,4] and in reports from grassroots organizations [5]. Moreover, healthcare professionals have been identified as a major source of discrimination [6], while at the same time biomedical research focuses mainly on reductionist approaches, most notably the minority stress model [7].

Minority stress was originally designed to account for the impact of discrimination based on sexual orientation in a way that was easily digestible for biomedical ideological frameworks [8]. The minority stress model developed by Meyer is based on an engineering analogy of stress, where all people live under the loads of individual and social stressors, with minority stress conceptualized as a third source of stress specific to non-heterosexual people [7]. Based on its effectiveness in understanding the mental health issues of non-heterosexual people, this model was transposed directly to TGD persons [9]. TGD people’s specific situation led to further developments and critiques of the model, which became independent and adopted as the gender minority stress model [8,9]. The central critique concerns the inadequacy of conceptualizing TGD stress as a variable independent of common stressors, or even integrating it solely through the prism of the individual integration of social norms [9]. Contributions from community and TGD health research have led to a reconceptualization of stress in the gender minority model as a consequence of the social environment in which TGD people exist [10]. Here, minority stress stems from the effect of social norms and ideologies on people’s social environment, impacting on their health, including their mental health [8,9]. In practice, the two have become autonomous models, and advances in the gender minority stress model have led to a global epistemological rethinking of how to integrate social norms into social practices and environments, raising the issue of their integration into research frameworks and practices [8].

A growing body of health research is incorporating normative elements from queer theories, in particular questioning social norms that shape our social practices: cis-normativity, hetero-normativity, binary genderism, or gender-normativity [8,11]—respectively, the norm of cisgenderism [12], heterosexuality, gender binarity, or gender experience (for more details, see [11]). While the latter provides a better understanding of TGD people’s oppression within social structures, they were in turn criticized by transmaterialist approaches stemming from transfeminism as abstract and disconnected from TGD people’s reality, and hence from means of intervention [13]. Building on Stone’s work [14], transmaterialism proposes a reading centered on the practical constraints in which TGD people evolve [13]. It is no longer a question of understanding how stress factors add up, or how the multiplication of transgressions to norms creates oppression, but how these factors and transgressions intersect to create radically different conditions of existence from which oppressions arise. Today, the multiplicity of models stemming from diverse scientific fields contributes to our difficulty in tackling these issues from a research stance [8], where it is noted that TGD people are generally rendered invisible in research productions and methodologies [15,16]. Moreover, the scientific fields coupled with the most discriminatory disciplines and practices produce the least effective explanatory models. There is therefore a real and practical need to improve biomedical understanding of vulnerabilities and oppressions experienced by TGD people, through the development and generalization of transdisciplinary approaches.

On the opportunity of the eleventh revision of the International Classification of Diseases (ICD-11), the World Health Organization (WHO) proceeded with what has been characterized as a depsychopathologized pathologization of TGD persons through the nosographic concept of gender incongruence [17,18]. This concept follows the cisnormative narrative inherited from the early 19th century [19], albeit its normative nature, i.e., that some people do not have a gender identity congruent with their assigned category at birth (ACAB). This discrepancy between perception and attribution would justify medical intervention, and medical authority would reinforce social norms in a world of slightly greater tolerance for TGD persons. The WHO based its reform on international field studies opposing its nosographic construction of gender incongruence to gender dysphoria [20]. Such narrow methodological framing highlights the temporal gap between humanities and nosographic research, whose intrinsically normative nature should put it in the avant-garde.

One of these field studies, carried out in France [21], claimed a participatory dimension by including TGD researchers in its methodology, and by carrying out the study at the Lille *Maison Dispersée de Santé* (MDS), a national landmark that pioneered an approach focused on harm reduction rather than on an intrinsically normative care pathway [18,22]. This involvement resulted in an additional questionnaire designed in collaboration with TGD members and specifically addressing their concerns [23,24]. However, while the existence of this questionnaire was instrumental in providing access to field knowledge and resources, its exploitation was never intended by the core—and cisgender—research team [23], limiting its existence to a doubly political argument: inside the study to gain access to the communities, and outside to artificially increase the validity of a nosographic modification from which TGD people remained effectively excluded [25]. Either way, this dataset presents both a rare opportunity to explore the discursive dimension of gender in TGD people, and a landmark in depsychopathologizing transness.

Data access was particularly challenging, involving nearly two years of procedures, even though the first author was a member of the study’s steering committee. Its analysis has already been the subject of two scientific publications from a trans perspective: one designed to contribute to public policy by reporting direct recommendations from TGD participants for promoting their health [26], the other investigating discriminatory biases in institutional and research processes [27]. These studies highlighted how the initial publication invisibilized a large part of the participants’ gender diversity by essentializing their gender identity within binary norms consistent with gender incongruence [26,27]. Thus, out of the 72 participants, the number of people existing outside binary genderism was reduced from 15—based on self-reported gender identity [26]—to 3 branded “queer” and removed from analyses [21] (see [15] for recommendations and examples of improved research practices).

It has been suggested that a better integration of sex and gender distinction would contribute to the validity of public health studies and improve our understanding of reality [1,15]. This paper therefore proposes to apply a trans and queer approach by focusing on a recurrent theme in health discrimination research: misgendering. Misgendering is an umbrella concept admitting several definitions and levels of application. For the purposes of this article, we will focus on the community definition of individual misgendering, i.e., not correctly gendering a person in a social interaction.

Rather than an explicit contribution to trans or queer studies, this paper focuses on queering [28] a pre-existing study. Its main objective is to contribute to enriching the discussion on better consideration of gender diversity in routine biomedical research by distancing the notion of misgendering from simplistic models based on gender identity—see, for example, [29]. As misgendering is a discursive practice, this study moves away from gender identity to look at how people talk and feel it is appropriate to talk about themselves. In so doing, it draws on a conception of language as a social practice, and of discourse as inscribed in a social context with which it interacts through processes of production and interpretation [30]. While the methodology applied is not, strictly speaking, critical discourse studies, it fits into a broader dynamic of integrating this field’s contributions into public health research [31].

Its main objectives are to account for gender diversity within a community of TGD people previously considered 95.8% binary [21], and to illustrate the possibility of norm-critical approaches within public health research. Using variables from the study’s main questionnaire, secondary objectives are to account for the interaction-specific nature of misgendering and the impact of the following: people’s requests to be gendered correctly, experiences of rejection and violence, and sexual orientation.

## 2. Materials and Methods

### 2.1. Epistemic Challenges

This paper draws on the first author’s experiential knowledge to reflect a trans and queer perspective. Indeed, it has been argued that the reproduction of normative frameworks in research stems notably from alterity between researchers and their research objects, leading to the development of new perspectives in trans and queer studies [16]. In this respect, this paper has a participatory dimension since it involves experiential knowledge in the subjectivity intrinsic to all research. It is also political, since these experiential knowledges can also be conceptualized as subjugated knowledge [32] in a minority context, and their revelation aims to contribute to changing research frameworks towards greater inclusivity and a better understanding of the complexity of the living world. Finally, it ties in with the depsychopathologization of trans persons and their experiences facing psychiatric medical authority—conceptualized and experienced as oppressive [17]. In all respects, it fits in with approaches developed by survivor research, which can attest to their greater relevance and ecological validity [33,34].

### 2.2. Ethical Conformity

The protocol complied with the Helsinki Declaration of 1975—as revised in 2008—and the WHO Good Clinical Research Practice guidelines. The research received approval from French biomedical research authorities: the *Comité de Protection des Personnes Nord-Ouest IV*, the *Comité Consultatif sur le traitement de l’information en matière de Recherche dans le domaine de la Santé*, and the *Commission Nationale de l’Informatique et des Libertés*. Informed written consent was obtained from every participant.

Data access was provided through a free of charge convention with the center.

### 2.3. Materials

Data were extracted from the 2017 study conducted at MDS in collaboration with WHO. Seventy-two transgender individuals receiving care at MDS voluntarily participated in the study. They were interviewed using two questionnaires: the main questionnaire translated and adapted from the one used in the Mexican study [35], and an additional one specific to the French study [21].

Primary data were obtained from five questions in the additional questionnaire: “during your transition, what word(s) did: you use most to refer to yourself? (QA1)/healthcare professionals use most to refer to you? (QA2)/your family or friends use most to refer to you? (QA3)/others use most to refer to you? (QA4)”; “Which of these words do you think would be most appropriate to refer to yourself and people transitioning? (QA5)”.

Remaining data were obtained from the main questionnaire. ACAB and gender identity were collected using a two-step method [36]. The questionnaire provided three modalities for the ACAB variable (female, intersex, and male) and five for the gender identity variable (genderqueer, intersex, man, transgender man/woman, and woman). Both questions also allowed for free responses.

Data for secondary analyses were also obtained from the main questionnaire: request to be gendered differently: overall/specifically by family and friends (binary yes/no); experiences of rejection and violence: overall/specifically by family and friends (binary yes/no); sexual attraction (men, women, both, none, and free answers).

### 2.4. Analyses

We first separated terms used to designate persons from other terms (notably derogatory, i.e., terms used as direct or micro-aggressions [37]) among answers to questions QA1–5. Derogatory terms were classified. Terms used to designate persons were categorized according to their grammatical gender and usage in context. The French language only features two genders (feminine and masculine), but an increasing number of expressions and emerging pronouns are being used to express neutral or other genders [38,39]. On this basis, data were categorized by the first author into feminine, masculine and neutral genders. Data from questions QA1 and QA5 were used as a basis for misgendering analyses among responses to questions QA2–4, each representing a group.

Results were arranged as co-occurrence networks to capture possible discrepancies between ACAB, gender identity, and discursive gender both employed by the individual and by healthcare professionals, family and friends, and others. Co-occurrence networks were created using Cytoscape v3.9.1. Based on graph theory, this software enables the analysis and visualization of complex networks.

For secondary objectives, we performed comparisons, based on the chi-square test, with the following hypotheses:-Individual misgendering is not group-specific;-Requesting to be gendered differently has an impact on occurrences of misgendering;-Misgendering is a form of aggression linked to rejection or violence;-Misgendering is not linked to sexual orientation.

During analyses, we clustered certain variables as follows: accepted vs. marginalized sexual orientation and binary vs. non-binary gender discourse practice (using one gender vs. more than one). All statistical analyses were performed using R v4.2.2. Descriptive statistics included frequencies (in percentages) for categorical variables, means and range for continuous variables.

## 3. Results

Table 1 summarizes the characteristics of participants, including the differences between the categories used further in the analyses.

All participants (n = 72; 100.0%) completed the first questionnaire, as well as the additional questionnaire questions on terms used by oneself (QA1), by family and friends (QA2), and appropriate to talk about them (QA5). Response rates for other questions remained high, covering terms used by healthcare professionals (QA2, n = 70; 97.2%), and others (QA4, n = 68; 94.4%). Participants were able to provide multiple responses (mean; range) for: QA1 (1.5; 1–4), QA2 (1.8; 1–5), QA3 (2.2; 1–7), QA4 (1.9; 1–11), and QA5 (1.4; 1–4). No participant reported only derogatory terms at QA1 and QA5, unlike QA2 (3; 4.3%), QA3 (1; 1.4%), and QA4 (1; 1.5%).

When asked about terms commonly used to designate them, 20 (27.8%) participants reported derogatory terms: 13 (18.6%) by their healthcare professionals, 5 by family and friends (6.9%), and 8 (11.8%) by others. Answers contained 35 derogatory terms: 15 (42.9%) by both healthcare professionals and others, and 5 (14.3%) by family or friends. They were grouped into three categories: psychopathologizing (15; 42.9%), transphobic (12; 34.3%), and miscellaneous (8; 22.9%). The psychopathologizing category mainly reported transsexual (11; 73.3%) and originated primarily from health professionals (13; 86.7%). The transphobic category mainly used insults derived from transvestite (8; 66.7%) and the pronoun it (2; 16.7%) and originated primarily from others (9; 75.0%). The miscellaneous category included homophobic insults (4; 50.0%) and other insults not detailed in the data (3; 37.5%) and originated mainly from others and not healthcare professionals (6; 75.0%).

With some participants reporting only derogatory terms in their responses, following their exclusion, the misgendering analysis covered 67 (93.1%) participants for QA2 and QA4, and 71 (98.6%) for QA3. Terms were categorized according to their grammatical gender and usage in context into three groups: feminine (F), masculine (M), and neutral (N). The neutral category was intended to capture people who “use neutral turns of phrase” (participant) or neologisms to distinguish themselves from traditional French uses of masculine and feminine. Each response was assigned to a category. Those categories were used to categorize participants. The possibility of multiple responses led each participant to be characterized using 1 to 3 for each question. For 27 (37.5%) participants, both appropriate and used terms did not necessarily match in practice. Figure 1 details these relationships.

For further analysis, we constructed a conceptual variable *pronouns*, which grouped together all categories used and appropriate for each participant. On this basis, answers to questions QA2, QA3, and QA4 that presented at least one different category were classified as misgendering, thus positioning ourselves in a hypothesis that doubly minimizes the occurrences of misgendering since it does not account for usage contexts—which the data did not include—and where multiple categories in the pronouns variable mathematically reduce the possibilities of identifying misgendering. Relationships with misgendering for each group, as well as results for ACAB and gender identity, are shown in Figure 2.

We found a statistically significant association between misgendering by family or friends and by others (χ^2^(1) = 4.3, *p* < 0.04), without this being found between misgendering by family or friends and by healthcare professionals (χ^2^(1) = 0.0, *p* > 0.9), or between misgendering by healthcare professionals and by others (χ^2^(1) = 1.6, *p* > 0.2). In addition, 9 (13.4%) participants reported misgendering by all three groups, including three who also reported derogatory terms: two (66.7%) by others, and one (33.3%) by their healthcare professionals. Only one (1.5%) participant reported derogatory terms from all three groups, combined with misgendering by family or friends. Figure 3 summarizes co-occurrences of misgendering and derogatory terms.

Having signified to other people a desire to be gendered correctly was not significantly associated with reporting misgendering by at least one of the groups (χ^2^(1) = 0.0, *p* > 0.9), or by others (χ^2^(1) = 0.0, *p* > 0.8). Similarly, misgendering by family or friends does not seem to be related to the explicit demand to be gendered correctly by them (χ^2^(1) = 0.1, *p*> 0.7). Nor was there a statistically significant link between misgendering by others and their rejection (χ^2^(1) = 0.1, *p* > 0.8) or having experienced violence (χ^2^(1) = 0.2, *p* > 0.6); nor between misgendering by family or friends and experience of rejection (χ^2^(1) = 1.6, *p* > 0.2) or violence (χ^2^(1) = 0.0, *p* > 0.9) by family or friends.

All participants answered about their sexual orientation, which we classified as accepted (legally recognized, n = 32; 44.4%), and marginalized (40; 55.6%) sexual orientations. The first group comprised 12 (16.7%) androsexual women, 8 (11.1%) gynosexual men, 7 (9.7%) gynosexual women, 2 (2.8%) androsexual transgender women, 1 (1.4%) androsexual man, 1 (1.4%) androsexual transgender man, and 1 (1.4%) gynosexual transgender woman. The second group comprised 17 (23.6%) bisexual persons, 16 (22.2%) pansexual persons, 6 (8.3%) asexual persons, and 1 (1.4%) non-binary androsexual person. Marginalized sexual orientations were significantly associated with being misgendered by others (χ^2^(1) = 3.8, *p* = 0.05), but not by health professionals (χ^2^(1) = 0.5, *p* = 0.5) or family and friends (χ^2^(1) = 0.8, *p* > 0.3). The small sample sizes did not allow for correlation analysis, but participants with marginalized sexual orientations faced more occurrences of derogatory terms by healthcare professionals (3 vs. 10; 9.4% vs. 25.0%), but not by family and friends (2 vs. 3; 6.3% vs. 7.5%) or others (4 vs. 4; 12.5% vs. 10.0%).

Building on this meta-categorization, we analyzed misgendering and derogatory terms experienced by organizing participants into two groups: those with single and binary modalities for the *pronouns* variable (binary pronouns: F or M; n = 36; 50.0%), and others (non-binary pronouns: FM, FN, FMN, MN, and N; n = 36; 50.0%). Results are presented in Table 2, and their reading should consider our approach, which doubly minimizes misgendering for those with more than one pronoun modality (i.e., 31 (86.1%) of the non-binary pronouns group).

The two groups defined by the pronouns variable had different proportions of participants declaring an accepted sexual orientation: 20 (55.6%) for binary pronouns, and 12 (33.3%) for non-binary pronouns; although this pattern is slightly above the threshold of statistical significance (χ^2^(1) = 3.6, *p* < 0.06).

## 4. Discussion

This paper admits several limitations. Firstly, this study consists of post hoc analyses using data collected in research conceptualized within a pathologizing discriminatory framework [21,23]. The data were collected to shed light on the WHO’s decision to maintain a depsychopathologized pathologization of TGD people, integrating transphobia directly into research conceptualization, in opposition to human rights-based approaches [40] which nonetheless constitute the WHO’s theoretical framework of reference. While the present study aims to explore the integration of social norms into research methodologies, it contains an inherent transphobic bias likely to minimize the results in relation to the reality of TGD people’s lives [21,23]. Secondly, the lack of a contextual element in the data has the effect of minimizing occurrences of misgendering for people discursively using several genders—the analysis methodology looks at whether the gender used is part of the participant’s reference genders and not whether it is the appropriate gender in the context of the interaction—and did not accurately allow participants with shifting gender identities to be accounted for. Finally, the questions did not specifically look for experiences of misgendering or derogatory terms and only captured those reported spontaneously, suggesting even more frequent occurrences in practice. Above all, this study took place in a normative context, where TGD people were interviewed in a care setting by a cisgender person—the initial proposal for a TGD interviewer having been rejected by the ethics committee [23,24]—and where participants were aware of the underlying political stakes of depsychopathologization. All these limitations lead us to consider this paper’s results, in terms of both frequency of occurrence and diversity, to be underestimated. While valid, the use of inferential statistics on a small, non-representative sample quickly reaches its limits in terms of generalization. Nonetheless, these methods were used in the first publication that contributed to the argument for depsychopathologization by the WHO [21]. Furthermore, our results pertain only to secondary objectives and are clearly polarized. Rather than aiming for generalization, they should be interpreted as a reflection of the material conditions and intersections of oppression experienced by some of the TGD population. This reinforces the public health relevance of adopting a materialist perspective over exploring gender solely as an explanatory variable.

One of the paper’s strengths is to draw upon extensive knowledge of discourse practices in TGD communities in continental France. As a language, French does not inherently allow for neutral personal expressions. This restriction is not disconnected from the historical context of its evolution and particularly normative nature [41]. These language-political stakes remain relevant today, to such an extent that the terms man and woman are currently used in French law to exclude TGD folks [42], and the neo-pronoun *iel* (French equivalent of *they*) faces political and institutional opposition [39]. Beyond gender grammar alone, access to community knowledge is necessary to identify expressions of gender neutrality and fluidity.

Another key strength of this study is its restriction of normative frameworks to those intrinsic to Western social norms: differential treatment between sexualities perceived as straight/homosexual and others, and binary genderism. No other a priori categories were employed in the analyses, which contributed to highlighting significant variability in participants’ genders. Our results reveal a significant number of disconnections between stated gender identity, used and appropriate gender. They allow us to move away from a standardized vision in which these elements are necessarily congruent, and to question their respective contexts. Gender identity is a social categorization bearing a political dimension that can vary according to social situations, just as the gender used also carries an intimate dimension linked to personal history, whereas appropriate genders reflect more a spectrum and individual language situations. By way of example, the first author is a transfeminine person who also defines herself as a transgender woman or as a non-binary person, depending on the context, and who employs she/her and they/them pronouns while overwhelmingly using the feminine to talk about themself.

If these variations reflect the incompleteness of our conceptual frameworks for accounting for gender experience in research, they also raise serious concerns over the reproduction of normative frameworks, given that the same sample of participants was a posteriori categorized by the researchers into 69 binary transgender people and 3 queer people [21]—compared with 15 based on declared gender identity [26] and 36 on the discursive used or appropriate gender.

Indeed, this discrepancy between participants existing within and outside the dominant trans narrative shapes the vast majority of the results, with a higher proportion of misgendering and use of derogatory terms. The main conclusion to be drawn here seems to be that the central factor in negative experiences is derogation from social norms shaping material conditions of existence, whether related to gender or sexual orientation—notions that appear linked.

Data collection on gender diversity is a current research issue, as is its identification in existing data showing similar trends [43], leading to the production of recommendations on how to construct structured data [44]. The invisibilization of non-binary or intersex people in published research is a central limiting factor to the production of knowledge for their health (see [45] for an example of implications in literature review methodologies). However, while inclusive data structuring can improve the participation of target audiences in research, it leads to the normative paradox of pushing back exclusion and invisibilization without solving the initial problem [46]. Rather than questioning whether gender should be collected in two or three steps (collecting ACAB and identity [36] or even sense of community belonging [47]), these results call for integrating its discursive dimension into health studies, including among non-TGD people. Indeed, beyond self-categorization, it is, above all, our interactional experience with others that will take precedence. This perspective fits in with current research dynamics, where natural language processing methods are being developed as tools for identifying gender minorities in data used in research [48]. Intrinsic knowledge of language usage in context is becoming a necessary component in the improvement of identification algorithms coupled with the collection of structured data [49]. On a broader level, variable combinations can be used to define existence groups within a given social context [13], and these conditions should form the basis of health analyses (see [50] for an example of categorization of TGD existences in contemporary French society). Reducing misgendering to a phenomenon exclusively specific to TGD people erases the experience of misgendering by non-TGD people, which is nonetheless recurrent and part of everyday experiences, including in the healthcare system.

Moreover, results clearly show that interactions with healthcare professionals are not safer or more appropriate than others. Therefore, if a transformation of systems is necessary, it should not only be based on developing cultural competence and tackling transphobia among professionals [51], but also on deconstructing our normative frameworks towards a generalized queering of our approach to others and to what we perceive as different or normal [11]. Otherwise, we will only progressively shift the problem, leaving the most vulnerable populations on the margins, with even fewer means at their disposal to assert their right to health.

## 5. Conclusions

In conclusion, a health promotion approach is part of a social justice perspective referring to the normative framework of human rights as defined since 1948 [52]. Gender identity and expression are fundamental human rights whose existence need not be restricted by social normative frameworks [53,54], themselves linked to systems of domination. If a better understanding of gender’s impact on health and an improvement of research processes are real objectives, one solution would be building on queer and trans studies advances. This study illustrates how these conceptual changes can be translated into a simple and adaptable methodological approach, including for studies initially built within restrictive normative frameworks.

## Figures and Tables

**Figure 1 ijerph-22-00178-f001:**
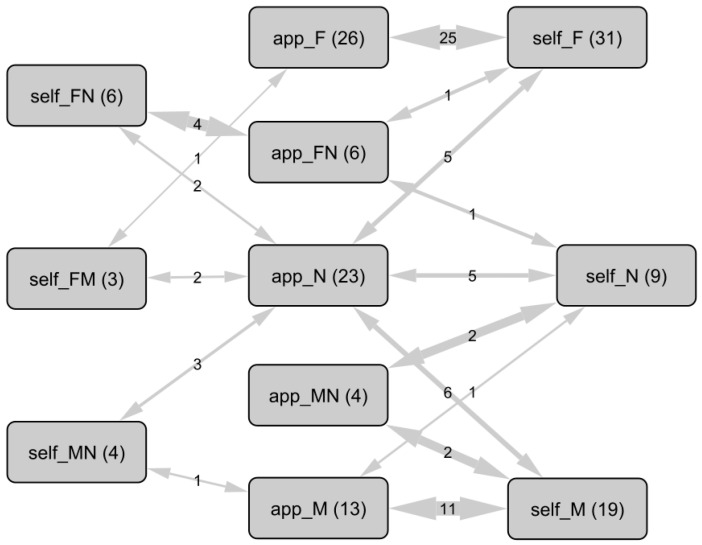
Network of relationships between categories of appropriate (app_) and used (self_) terms to talk about oneself (n = 72). Nodes represent categories (participants), edges represent co-occurrences and their respective counts. Modalities: feminine (F), masculine (M), and neutral (N). For example, among the 13 people stating that the masculine pronoun is the appropriate one to refer to them, 11 use it exclusively, 1 person uses neutral forms, and 1 person uses both.

**Figure 2 ijerph-22-00178-f002:**
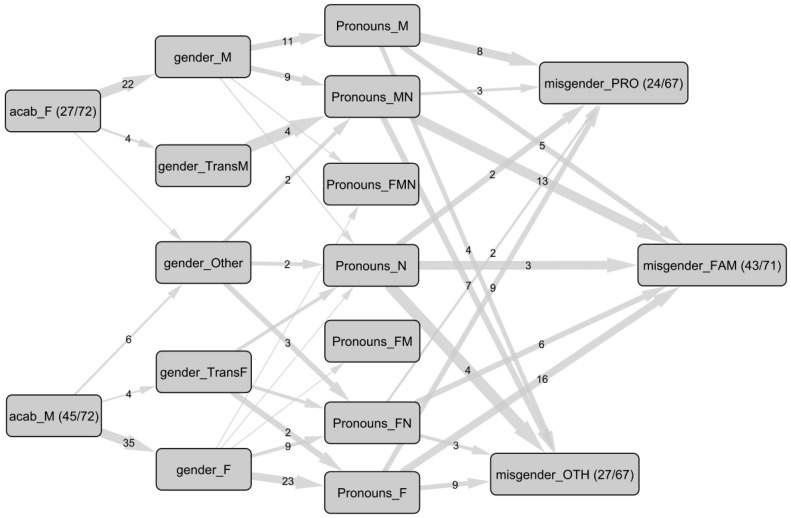
Network of relationships between assigned category at birth (acab_), gender identity (gender_), pronouns, and misgendering (misgender_). Nodes represent categories (participants/total), edges represent co-occurrences and their number. Lack of number on an edge means a single account. Modalities: Acab: female (F), and male (M); Gender: woman (F), transgender woman (TransF), man (M), transgender man (TransM), and other; Pronouns: feminine (F), masculine (M), and neutral (N); Misgender: by health professionals (PRO), by family or friends (FAM), and by others (OTH). For example, among the 45 people assigned to the male category at birth, 35 identify with a feminine gender identity; 23 of them use exclusively feminine pronouns, and 9 of them are misgendered by their healthcare professionals among the 24 individuals in the sample who experience misgendering.

**Figure 3 ijerph-22-00178-f003:**
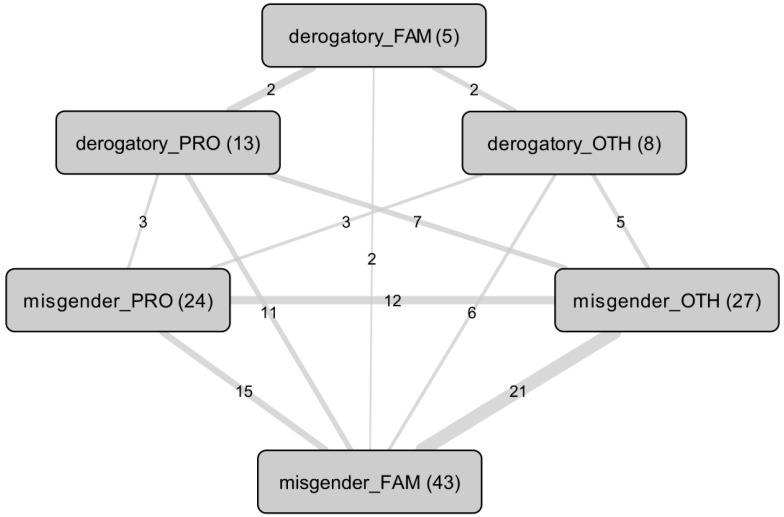
Network of co-occurrences between misgendering (misgender) and derogatory terms (derogatory). Nodes represent categories (participants), edges represent co-occurrences and their respective counts. Modalities: by health professionals (PRO), by family or friends (FAM), and by others (OTH). For example, among the 43 people misgendered by their family, 15 are also misgendered by healthcare professionals, 21 by others, 11 are referred to with derogatory terms by healthcare professionals, 2 by their family, and 6 by others.

**Table 1 ijerph-22-00178-t001:** Description of participants.

	Total (*n* = 72)	Pronouns in Discourse		Sexual Orientation	
		*Binary* (*n* = 36; 50.0%)	*Non-Binary* (*n* = 36; 50.0%)	*Accepted* (*n* = 32; 44.4%)	*Marginalized* (*n* = 40; 55.6%)
**Assigned Category At Birth** (*F*/*M*)	27 (37.5%)/ 45 (62.5%)	11 (30.6%)/ 25 (69.4%)	16 (44.4%)/ 20 (55.5%)	10 (31.3%)/ 22 (68.8%)	17 (42.5%)/ 23 (57.5%)
**Gender Identity**					
*Man*	22 (30.6%)	11 (30.6%) **	11 (30.6%) **	9 (28.1%)	13 (32.5%)
*Other*	15 (20.8%)	2 (5.6%) **	13 (36.1%) **	4 (12.5%)	10 (25.0%)
*Woman*	35 (48.6%)	23 (63.9%) **	12 (33.3%) **	19 (59.4%)	17 (42.5%)
**Age** (mean; range)	28.0; 18–50	29.7; 18–50	26.3; 18–48	31.2; 18–50	25.4; 18–48
**Monthly income** in euros (*n* = 71; mean; range)	995.9; 0–6000	961.2; 0–3000	1031.7; 0–6000	1117.1; 0–6000	896.5; 0–4000

** χ^2^(2) = 11.5, *p* < 0.004.

**Table 2 ijerph-22-00178-t002:** Distribution of usage of misgendering and derogatory terms, by group and pronoun binarity, for all participants (n = 72; 36 binary).

Group	Misgendering			Derogatory Terms		
	Health Professionals	Family and Friends	Others	Health Professionals	Family and Friends	Others
(*n*; *n[b]*/*n[nb]*) ^a^	(67; 33/34)	(71; 36/35)	(67; 35/32)	(67; 33/34)	(71; 36/35)	(67; 35/32)
**Binary pronouns** [*F*; *M*] ^b^	17 * (70.8%)	21 (48.8%)	13 (48.1%)	3 (23.1%)	0 (0.0%)	3 (37.5%)
**Non-binary pronouns** [*FM*; *FN*; *FMN*; *MN*; *N*] ^b^	7 * (29.2%)	22 (51.2%)	14 (51.9%)	10 (76.9%)	5 (100.0%)	5 (62.5%)
**Total events**	24 (35.8%)	43 (60.6%)	27 (40.3%)	13 (19.4%)	5 (7.0%)	8 (11.9%)

^a^ total answers (*n*); participants using binary (*n[b]*) and non-binary (*n[nb]*) pronouns. ^b^ reference pronouns: feminine (*F*), masculine (*M*), neutral (*N*). * χ^2^(1) = 7.0, *p* < 0.009.

## Data Availability

Data were accessed by a convention with the World Health Organization Centre for research and training in mental health (*EPSM Lille-Métropole*) which does not allow public release of materials. Original study data can be accessed by requesting the owner.

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
