# Peer review of "A Trans and Queer Discursive Approach to Gender Diversity and Misgendering in the Transgender and Gender Diverse Population: Queering a Study for ICD-11"

_ijerph, 2025, doi:10.3390/ijerph22020178_

Round 1

Reviewer 1 Report (Previous Reviewer 3)

Comments and Suggestions for Authors

The revisions comprehensively addressed my comments.

Author Response

We would like to thank you for your positive feedback and the improvements made possible by your review.

Reviewer 2 Report (Previous Reviewer 2)

Comments and Suggestions for Authors

1. Lines 309-315 remain unclear. 

2. It would be helpful if the authors particularized the relevance of this research in terms of future trans studies. 

Author Response

Many thanks for your feedback, we have made the following changes in response:

  1. Comment 1: Lines 309-315 remain unclear.
  • Response 1: We reworded the first two limitations identified in our study, which were presented on lines 310-315. Lines 324-337 now set out data-related biases in a structured manner. We have developed the biases inherent in the design of data collection, and improved the wording as well as developed the second limitation on the undermining of results inherent in our methodology and the absence of contextual data on the interaction. We hope these changes will make the text clearer and the limitations more apparent.
  1. Comment 2: It would be helpful if the authors particularized the relevance of this research in terms of future trans studies.
  • Response 2: We have developed this aspect at length in lines 389-405. To this end, we have added references to similar trends, current issues in data collection and its limitations, its impact on current knowledge illustrated by a literature review, and integrated it into current perspectives on the use of textual data for the identification of gender minorities.

We would like to thank you for your review, which helped us to improve the understandability of this research.

Reviewer 3 Report (Previous Reviewer 1)

Comments and Suggestions for Authors

This trans research illustrates methodological challenges through queering an earlier study by focusing on misgendering as a discursive element. Authors based our work on discursive materials reported by TGD participants in an ICD-11 study on gender incongruence. The authors used network analyses to illustrate potential differences between declared gender identity and discourse practices. Results highlight a gap 18 between declared gender identity and discourse practices, bringing the number of gender-diverse non-binary participants in the sample from 15 (20.8%) to 36 (50.0%). Moreover, misgendering and the use of derogatory terms are more common toward gender-diverse individuals. Sexual orientation shows a similar trend. This study reveals the reproduction of social norms within research processes and medical knowledge, as well as how, from an individual perspective, their non-compliance seems to be a key factor in TGD individuals' experience. By providing this simple methodological example, the authors hope to promote better integration of gender and its various dimensions into biomedical and public health research.

Overall the authors have addressed some of the previous feedback provided.  That being said, I think the authors would benefit from stepping back and focusing on a few key points below such as the use of minority stress theory in the referenced context of sexual orientation and the critical lack of the important connections to transgender theory.  Please see below

Overall feedback

p. 1 The authors have now moved to minority stress theory in the context of sexual orientation but have conflated sexual orientation and gender identity. The authors should rethink the use of minority stress theory.  Again, there are important transgender theoretical works in the field that focus on social norms and social practices.  Please cite some of this research and make some of these important connections.

p. 2 I also question why there is a discussion of pathologization without connecting directly to transphobia (Hill & Willouby, Nagoshi et al,) etc. as the former manuscript did.    

p. 9 The research states “these are post-hoc analyses of a study designed to reinforce the dominant trans narrative in a pathologizing context.” It seems that the authors cherry picked a study.  This research would have benefited though from more meta- analyses of post hoc studies, therefor lacking generalization.    Can the authors add two more sources of data collected in order to show a meta trend with a total of 4 citations?

Author Response

Overall the authors have addressed some of the previous feedback provided.  That being said, I think the authors would benefit from stepping back and focusing on a few key points below such as the use of minority stress theory in the referenced context of sexual orientation and the critical lack of the important connections to transgender theory.  Please see below

  • We would like to thank you for your careful review, we have addressed your overall feedback to the best of our ability:

Overall feedback

  1. Comment 1: The authors have now moved to minority stress theory in the context of sexual orientation but have conflated sexual orientation and gender identity. The authors should rethink the use of minority stress theory. Again, there are important transgender theoretical works in the field that focus on social norms and social practices.  Please cite some of this research and make some of these important connections.
  • Response 1: Thank you for your comments. From a meta point of view, this section's challenge was to clearly argue the relevance of normative issues in scientific productions. Clearly, our modifications were too superficial, as you pointed out. We stuck to the minority stress model as it is more operative than discrimination and we feel contributes to the readership's accessibility and consideration of this issue.
  • Instead of citing it as an entry point, we've devoted an entire introductory paragraph (lines 38-58) to the evolution of the minority stress model for TGD people and its relationship to social normative issues. We have added 3 references: one on a specific model of gender minority stress and two critical analyses of concepts.
  • We thank you for your vigilance and believe that the current form, although more developed than we had anticipated in the initial writing, helps to remove ambiguity and facilitate the integration of normative issues in the paper.

  1. Comment 2: I also question why there is a discussion of pathologization without connecting directly to transphobia (Hill & Willouby, Nagoshi et al,) etc. as the former manuscript did.
  • Response 2: This is a writing artifact, and we have reintroduced the notion of transphobia at the start of the discussion (l324-329) with regard to the WHO approach, and added a reference to a human rights-based approach to nosography. We preferred this breakdown, where the entry via the minority stress model tackles issues with more immediate ramifications, and where the discussion develops issues of discrimination in research and health systems.

  1. Comment 3: The research states “these are post-hoc analyses of a study designed to reinforce the dominant trans narrative in a pathologizing context.” It seems that the authors cherry picked a study. This research would have benefited though from more meta- analyses of post hoc studies, therefor lacking generalization. Can the authors add two more sources of data collected in order to show a meta trend with a total of 4 citations?
  • Response 3: As we previously stated, this choice of data is not only symbolic in nature, but also reflects limitations in data access in view of our resources. Since it is not materially possible for us to integrate other data into the analysis, we have provided a detailed paragraph on integrating current research in lines 389-405.
  • In particular, we've added a study that discusses similar issues in reference to census data. However, despite further extensive research, we were unable to identify many studies similar to ours, which also confirms our willingness to continue this project in the future on the basis of your feedback.
  • To compensate, we have included a literature review on the subject which points to similar trends in its sample of publications. We've also included references on structured data collection, based on feedback from communities that have been left out of research, and developed the paradox of data structuring in a non-normative context. Subsequently, we have also extended the ramifications of our findings through use of natural language processing methods for the identification of gender-diverse individuals in the samples.
  • We believe these modifications will enable greater integration of our results into the existing literature and help to illustrate that, while this study is specific to a precise sample, its conclusions are in line with a trend in publications on the subject.

We would like to thank you warmly for your rigorous feedback, and hope that the changes we have made are sufficient and appropriate.

This manuscript is a resubmission of an earlier submission. The following is a list of the peer review reports and author responses from that submission.

Round 1

Reviewer 1 Report

Comments and Suggestions for Authors

This trans research illustrates methodological challenges through queering an earlier study by focusing on misgendering as a discursive element. They based their work on discursive materials reported by TGD participants in an ICD-11 study on gender incongruence. Results highlight a gap between declared gender identity and discourse practices. Overall the authors make an important note of how the positivistic, binary way of conducting and collecting research with TGD individuals has caused large amounts of misgendering.  The authors look to methodologies of misgendering as a way to support their arguments.  Overall the authors seem to conflate transphobia with a lack of exploration into cisgender norms or heteronormative ideologies; displaying a need for more research by the authors into important correlates of transphobia and theoretical developments of transgender theory to practice.  The authors make a great point about how misgendering draws on a conception of language as a social practice and of discourse as inscribed in a social context with which it interacts through processes of production and interpretation. This research would have benefited though from more meta-analyses of post hoc studies, therefor lacking generalization. Also the authors note 2-3 steps being taken to decrease misgendering without displaying how to do so.  Additional feedback is below.

p. 1 The authors note “ This marks a gap with other research fields, such as queer studies, where developing queer research ethics shifts the focus from increasing tolerance or cultural competence among professionals to reexamining social norms that shape our social practices.” There are numerous transgender theatrical works in the field that focus on social norms and social practices.  Can the authors cite some of this research and more clearly define the limitations of transgender theory?

 p. 1The authors state “Thus, it no longer revolves around reducing phobias, but rather towards examining cis-normativity, hetero-normativity, binary genderism, or gender-normativity – respectively, the norm of cisgenderism, heterosexuality, gender binarity, or gender experience” Have the authors explored other correlates of transphobia with heteronormativity, eg Nagoshi et al 2008?

p. 2 The authors state “Unlike vulnerable minority approaches, queer approaches are intended to benefit the whole population.”  Do the authors believe that theories of minority stress and transgender theories are not applied to larger populations as well?

p. 2 The authors note “…who face increased discrimination, social rejection and violence to an extent that cannot be captured by the notion of transphobia alone and to which invisibilization in scientific productions and research methodologies contributes.” Unclear where research on transphobia would be considered to be in a silo and not applicable to public health concerns.  Have the authors looked into the important connections that have been made between TGD research and broader population health? Eg. Galupo et al, Kattari et al

 p. 2 The authors also state “the number of people existing outside binary genderism was reduced from 15  based on participants' self-reported gender identity to 3 branded "queer" and moved from analyses” .  Can the authors note research that uses more current forms of self-reporting that exceeds the term queer that can be so limiting?

p. 8 The research states “these are post-hoc analyses of a study designed to reinforce the dominant trans narrative in a pathologizing context.” It seems that the authors cherry picked a study.  This research would have benefited though from more meta- analyses of post hoc studies, therefor lacking generalization.    Can the authors explore more than one source of data collected in order to show a meta trend?

p. 9 The authors note “Rather than questioning whether gender should be collected in two or three steps, these results call for integrating its discursive dimension into health studies, including among non-TGD people.”Can the authors offer more information on what that 2-3 step process would look like?  In terms of public health needs, there is still a gap here on how to fit the needs of the TGD community without basic demographic data being collected, note that this has been a long-term struggle with the term “other” in the demographics form, so then what do the researchers offer as a solution to this complex issue?

Reviewer 2 Report

Comments and Suggestions for Authors

1.     The value of this submission is unclear. What are the objectives of the researchers?

2.     The authors should acknowledge that many social scientists do not focus on constructs such as “homophobia” and “transphobia” (i.e., they recognize that the negativity directed against sexual and gender marginalized persons does not stem from a phobic response).  

3.     The authors should acknowledge that “vulnerable minority approaches” also benefit the majority.

4.     What do the authors mean by “derogatory terms” (see p. 4)?

5.     How precisely is misgendering being determined?

6.     The figures are unclear.

7.   What are the authors using the term “normal” to describe sexual orientation?

Reviewer 3 Report

Comments and Suggestions for Authors

Interesting paper to consider networks in increasing the identification of TGD individuals.

I had problems with the (1) presentation of arguments and (2) the use of statistics. With the presentation of arguments, I am missing explicit statements that should be in the main text. The abstract says that misgendering in networks increases the count of TGD individuals. That should be explicitly stated in the main text. The abstract says there is an increase of 15 to 36 TGD. I did not see that in the main text. How many people were in the sample? I found number 72 in the table, but that should be in the main text. Another presentation issue is in the results section, I was told by the authors that some TGD persons were misgendered, but I wished there was a detailed explanation in the analysis section on how the authors define misgendering in the sample.

I do not like the use of inferential statistics with a small sample size. I would rather see just descriptive statistics, and it would be adequate to me for the authors to show that there is misgendering with many people.

Comments on the Quality of English Language

Some of the language I am not familiar. What is MDS, marginalized sexual orientation, and discursive pronouns?
